



# Improved landscape partitioning and estimates of deep storage of soil organic carbon in the Zackenberg area (NE Greenland) using a geomorphological landform approach

Juri Palmtag[1], Stefanie Cable[2, 3], Hanne H. Christiansen[2, 3], Gustaf Hugelius[1] and Peter Kuhry[1]

[1] Department of Physical Geography, Stockholm University, 106 91 Stockholm, Sweden
[2] Center for Permafrost (CENPERM), Department of Geosciences and Natural Resource Management, University of Copenhagen, 1350 Copenhagen, Denmark
[3] Arctic Geology Department, The University Centre in Svalbard (UNIS), 9170 Longyearbyen, Norway

*Correspondence to*: Juri Palmtag (juri.palmtag@natgeo.su.se)

**Abstract.** This study aims to improve the previous soil organic carbon (SOC) and total nitrogen (TN) storage estimates for the Zackenberg area (NE Greenland) that were based on a
land cover classification (LCC) approach, by using geomorphological upscaling. In addition, novel SOC estimates for deeper deposits (down to 300 cm depth) are presented. We hypothesize that landforms will better represent the long-term slope and depositional processes that result in deep SOC burial in this type of mountain permafrost environments. The updated mean SOC storage for the 0–100 cm soil depth is 4.8 kg C m$^{-2}$, which is 42% lower than the
previous estimate of 8.3 kg C m$^{-2}$ based on land cover upscaling. Similarly, the mean soil TN storage in the 0–100 cm depth decreased with 44% from 0.50 kg (± 0.1 CI) to 0.28 (± 0.1 CI) kg TN m$^{-2}$. We ascribe the difference to a previous areal overestimate of SOC and TN-rich vegetated land cover classes. The landform-based approach more correctly constrains the depositional areas in alluvial fans and deltas with high SOC and TN storage. These are also
areas of deep carbon storage with an additional 2.4 kg C m$^{-2}$ in the 100–300 cm depth interval. This research emphasizes the need to consider geomorphology when assessing SOC pools in mountain permafrost landscapes.

## 1 Introduction

Permafrost soils in the northern circumpolar region are sensitive to climate change. In addition,
they store large amounts of soil organic carbon (SOC) that has accumulated under low ground temperatures over decadal to millennial timescales. In most regions, permafrost temperatures have increased since the 1980s and increased global mean surface temperatures are projected to decrease the near-surface permafrost extent by 37% to 81% (with RCP2.6 to RCP8.5, respectively) by the end of the 21st Century (IPCC 2013). The observed permafrost
degradation could intensify microbial activity and increase decomposition of organic matter formerly stored in permafrost, releasing more greenhouse gases into the atmosphere and



providing a positive feedback on global warming (Schuur et al., 2015). Over a decade ago, permafrost was identified as a major vulnerable terrestrial carbon pool (Gruber et al., 2004). The first assessment of the magnitude of the Northern Permafrost Region SOC pool was made by Tarnocai et al. (2009) who estimated it to be as large as ca. 1700 Petagram C (PgC). Since

then, availability of more data has constrained this estimate to ca. 1300±200 PgC (Hugelius et al., 2014). However, substantial uncertainties and data-gaps remain, particularly for high-Arctic and mountainous landscapes. This limited knowledge on the spatial distribution of SOC in permafrost landscapes remains a strong constraint on the ability to predict the vulnerability of the permafrost SOC pools from local to landscape to pan-Arctic scales.

Most landscape to regional-scale estimates of permafrost SOC stocks have used thematic maps to stratify and scale point observations to full spatial coverage (Hugelius, 2012). There are numerous examples of studies applying land cover classifications (LCC) for SOC upscaling in permafrost landscapes (e.g. Kuhry et al., 2002; Hugelius and Kuhry, 2009; Palmtag et al., 2015, 2016) In Palmtag et al. (2015), a LCC was applied for upscaling SOC for two lowland

sites in NE Siberia and a mountainous site, Zackenberg Valley, in NE-Greenland. However, limitations have been pointed out especially for the mountainous Zackenberg Valley site where the geomorphology and cryostratigraphy are highly heterogeneous (Cable et al., 2017). In this area, relatively shallow deposits occur on steeper slopes, while the foothills accumulate massive deposits over millennial time-scales, involving carbon burial with syngenetic

permafrost aggradation (Palmtag et al., 2015; Cable et al., 2017). Thus, the application of LCC-scaling for SOC upscaling in high relief landscapes can involve larger uncertainties, particularly when estimating deeper SOC of colluvial and alluvial deposits because the long-term depositional history, controlling SOC burial, cannot be captured by the current vegetation cover that is primarily reflected in LCC. Combining a geomorphology-based landscape

classification (GLC), now available for the Zackenberg Valley (Cable et al., 2017), with high quality data from detailed field studies may, therefore, improve SOC upscaling in high relief permafrost landscapes.

The overall aim of this study is to improve the SOC and total nitrogen (TN) storage estimates for the Zackenberg area (NE Greenland). Specific objectives are (1) to use largely the same

dataset as in Palmtag et al. (2015) to upscale SOC and TN at 0–100 cm depth to landscape scale, based on geomorphological mapping, (2) to compare the results with the previous LCC upscaling approach, (3) to present the first SOC estimates for deposits deeper than 1 m based on newly collected deep pedons, and (4) to evaluate the importance of geomorphology for assessing landscape level SOC storage.

## 2 Study area


The Zackenberg valley (c. 74°28'N, 20°34'W) is a mountainous high-arctic tundra area with a mean annual temperature of –9.2°C located within the continuous permafrost zone of NE Greenland (Fig. 1). The mean annual precipitation is 261 mm (Hansen et al., 2008). The study





area extends from sea level at the shores of the Young Sound up to 1372 m at the top of
Zackenberg Mountain. A large fault system divides the weathering resistant Caledonian
gneiss/granite bedrock in the West from the Cretaceous-Tertiary sedimentary rocks in the east
(Escher and Watt, 1976). According to Bennike et al. (2008), the valley deglaciated prior to 11

300 cal yr BP (calendar years Before Present). The parent material in the low-lying central
valley is dominated mostly by glacial, periglacial, alluvial, fluvial and deltaic deposits, while
on the slopes either boulder-fields or colluvial sediment predominate (Christiansen et al., 2008;
Cable et al., 2017). A weakly developed Typic Psammoturbel is the prevailing soil type of the
central valley, with Gelorthents on the slopes (terminology following Soil Survey Staff, 2014).

Small areas were occupied by peaty soils, mainly Histoturbels.

**3 Methods**

**3.1 Soil sampling**

Field work was conducted during late summer in 2009, 2012 and 2013. In 2009 and 2012, 38
sampling sites were selected using a semi-random, stratified transect sampling approach with

predefined equidistant pedon intervals of 100 to 500 m using a handheld GPS device. Mineral
soil samples were collected using a steel pipe, manually hammered into the soil. A more
detailed sampling procedure is described in Palmtag et al. (2015). In 2013, additional deeper
core pedons (down to 455 cm) were sampled by drilling in alluvial fans (Cable et al., 2017),
using a handheld motorized Earth Auger (STIHL BT 121). Out of the total of 48 sites used in

this study (Fig. 1), eight sites were sampled to depths of more than 200 cm, 19 sites to between
100 and 200 cm depth, and the remaining sites to less than 1m depth, primarily due to shallow
mineral soils overlying the bedrock. There were 648 samples collected in total throughout the
three field seasons consisting of on average 10 cm long increments of organic layer, active
layer and permafrost samples.

25  **3.2 Soil chemical analyses and SOC/TN calculations**

Each sample had a known field volume and was oven-dried, weighed and sieved to determine
the dry bulk density (DBD, g cm$^{-3}$) and the amount of coarse fragments (CF, >2mm, %).
Subsequently, each sample was individually homogenized and burned to obtain loss on ignition
(LOI; Heiri et al., 2001) at 550°C and, about every second sample, at 950°C to determine,

30  respectively, its organic matter and inorganic carbonate content through weight loss (for
details, see Palmtag et al., 2015).
To determine total organic carbon and nitrogen (TOC/TN, %), ca. 70% of the samples (all
samples from fieldwork in 2009) across all sites and horizons were analysed using an EA 1110
Elemental Analyzer (CE Instruments, Italy). To calculate the soil organic carbon content for

35  the remaining samples we used a fifth order polynomial regression ($R^2$= 0.97) between LOI at
550 °C and TOC values from the elemental analyser. This high order polynomial regression





was necessary to correctly represent %C at very low LOI 550 °C. Results for the inorganic carbon content were very low. The latter measurements were based on 297 samples with an average LOI 950 °C weight loss of 0.994%.

The SOC and TN storage (kg m$^{-2}$) was calculated for each collected sample using the available data on DBD (g cm$^{-3}$), %C or %TN, 1–CF (the remaining proportion of the sample after excluding the coarse mineral fractions (>2 mm)), thickness T (cm) of the sample, multiplied by 10 for unit conversion (see equation 1 and 2):

$$SOC = (DBD * \%C * (1 - CF) * T) * 10 \qquad (1)$$
$$TN = (DBD * \%TN * (1 - CF) * T) * 10 \qquad (2)$$

Then, the total SOC and TN (kg m$^{-2}$) storage was calculated by summing all the individual samples from the same profile to the different reference depths of 0–30 cm, 0–100 cm, 0–200 cm, etc. In cases when a sample was missing, the gap was interpolated from samples above and below by taking into account field notes on texture, ice content, coarse fraction and any presence of buried C-enriched soil layers. Due to data limitations, the TN storage was calculated for the 0–100 cm depth interval only.

In total, we used 48 sampling sites to estimate SOC storage. If bedrock was hit at any point (n=8 within 0–100 cm depth; n=10 within 100–200 cm depth; n=5 within 200–300 cm depth), we used a SOC content of 0 kg C m$^{-2}$ for the remaining bedrock part down to 300 cm depth. In the 25 remaining sites within the depth range of 0–100 cm, 100–200 cm and 200–300 cm, where bedrock was not reached but sampling was stopped because of time or logistical constraints, the SOC content was extrapolated to the full 100, 200 or 300 cm depth intervals by taking into account the location, topography, geomorphologic landform, sediment type and information from other similar sites.

### 3.3 Upscaling procedure

The upscaling from the field measurements to landscape scale was performed in ArcGIS 10.2 (ESRI, Redlands, CA, USA) by multiplying arithmetic means of SOC from all sites belonging to the same landform class with the extent of that same class in the digital geomorphology map at a 1:10 000 scale from Cable et al. (2017). The geomorphological map is based on geomorphological mapping using ortho-rectified panchromatic aerial images of 0.2 m resolution, and field validation. The 48 SOC sampling sites covered most, but not all of the originally recognized 28 landforms, however, 12 of these occupied only negligible areas of <3% of the total study area (Table 1). To achieve full coverage across the study area, the mapped landforms were merged into larger geomorphological classes based on topographic position and overall geomorphological characteristics. The adjusted map, consisting of 10 geomorphological classes, was then used for SOC upscaling. The map has a terrestrial coverage of 111 km$^2$ with an additional 17 km$^2$ of sea area (Young Sound/Tyrolerfjord), which is not included in the upscaling, but visible on the map (Fig. 1). The extent of this





geomorphological map nearly completely overlaps with that of the LCC map used by Palmtag et al. (2015).

The arithmetic mean SOC storage with standard deviations (SD) for different depth intervals in the active layer and permafrost (and TN 0–100 cm storage) were calculated for each landform 5 based on all study sites in each landform. One landform (bedrock) consists of only one sampling site but, since SOC in rock walls is considered negligible, any assumed small within-class variability barely affects the SOC estimate at landscape scale. Subsequently, the landscape mean SOC storage for the whole study area was calculated from the mean values of each geomorphological class multiplied with the proportion of the area occupied by that class 10 in the simplified geomorphological map.

**3.4 Statistical analyses**

To provide reasonable error estimates for landscape SOC, which vary naturally in the environment, a spatially weighted 95% confidence interval (CI) was calculated following Thompson (1992). This CI is calculated to account for the relative spatial coverage, storage 15 variability and degree of replication of each upscaling class using equation 3:

$$CI = t * \sqrt{\sum \left( \left( a_i{}^2 * SD_i{}^2 \right) / n_i \right)} \qquad (3)$$

Where 't' is the upper $\alpha/2$ of a normal distribution (1.96); 'a' the percentage of the total area 20 per class; 'SD' the standard deviation of the storage estimate per class; 'n' the number of replicates per class; and 'i' refers to the specific land form classes. The applied upscaling procedure assumes that the available sample is sufficiently replicated to accurately reflect the natural variability within a class (Hugelius, 2012). It is important to note that the presented CI ranges do not account for any spatial errors in landform upscaling products. Error estimates for 25 landscape SOC as well as the analyses of variances with means and SD were performed using the software MS Excel.

**4 Results**

The Zackenberg study area consists of several main geomorphological classes (Figs. 1 and 2). Exposed bedrock occupies about 8% of the study area, containing negligible SOC storage (Fig. 30 2 and Table 2). The most widespread landform class (30% of the study area) is 'allochthonous weathered bedrock' (n=2). This class is predominantly located at higher elevation on steep hillslopes and consist of coarse-grained colluvium deposits with very little soil and vegetation development, deposited by either downslope creep and/or gravity depositional sorting. Only 7% of the surface of this class is actually vegetated, leaving 93% bare ground and exposed 35 bedrock. This landform is rather active leading to relative shallow sediments with very low SOC content (Fig. 2 and Table 2).



The landform 'solifluction sheets' (n=4) covers 14% of the study area, is located on intermediate hillslopes and consists mainly of fine-grained colluvium deposits, in general loose unconsolidated weathered sediments deposited by slow downslope movement of water-saturated sediment due to recurrent freezing and thawing of the ground and driven by gravity.

Only 18% of the area is vegetated, with the remaining 82% occupied by boulders. SOC content is low (Fig. 2 and Table 2).

The class 'lateral and end moraines' (n=4) occupies 18% of the area. More than 60% of its surface area is occupied by boulders. This landform is inactive (not eroding or having sedimentation) but sparsely vegetated and consists largely of coarse diamictons with shallow

soil depths (<40 cm) and low SOC contents (Table 2). The 'ground moraine' class occupies about 2% of the study area in the lower parts of the central valley (Figs. 1 and 2). The surfaces have been stable since the early Holocene and are largely vegetated with only 1–5 % boulders. Soils have developed in the top meter, with signs of cryoturbation, leading to high SOC contents in the 0–100 cm depth interval (Table 2). Deposits below 100 cm depth in these two

glacially deposited classes are considered tills, with very low SOC contents (Fig. 2 and Table 2).

Another widespread landform (15% of the study area) is referred to as 'alluvial fans' (n=15), with additional small areas of peaty fens (n=5) and bogs (n=3) on alluvial fans (0.4%). These are areas of high SOC storage with on average 19.8, 29.8 and 42.7 kg C m$^{-2}$ for the top 100 cm,

respectively (Fig. 2 and Table 2). The alluvial fans are generally located in the foothills of the sedimentary northeastern slopes of the study area. Wetlands are common in the lowermost reaches of these foothills, where the slope decreases and water accumulates. Alluvial fan deposits have fine-grained sediments, often containing thin buried C-enriched layers, which sometimes reach depths of >300 cm (maximum observed depth of 370 cm). As a result, this is

the landform with some of the highest 100–200 and 200–300 cm SOC stocks (Fig. 2 and Table 2), contributing most to the deeper SOC storage in the study area. For calculation purposes bog hummocks (isolated palsas and pounus), with very high SOC stocks but occupying only 3% of the total wetland area, were separated from peaty fen areas to not overestimate total SOC storage in this class. Furthermore, wetlands (including bogs) on alluvial fans occupy only 0.4%

of the total study area and their high SOC storage increases the weighed mean SOC storage for the entire study area by only 0.1 kg C m$^{-2}$. The small area of freshwater lakes (<1%) has intermediate SOC values down to 300 cm of depth (Fig. 2 and Table 2).

Relict fluvial and (raised) deltaic landforms occupy about 4% of the study area, in the lower reaches of the central Zackenberg valley. These landform classes have high SOC values down

to depths of 300 cm and more, contributing significantly to the overall deep SOC storage in the study area. Recent fluvial stream beds occupy about 8% of the area, but have low SOC storage values at all considered soil depth intervals (Fig. 2 and Table 2).

For the entire study area, the estimated weighed mean SOC storage is 4.8 kg C m$^{-2}$ in the top 100 cm (Table 2). When comparing the mean SOC 0–100 cm distribution among different

layers, 13% is stored in the organic layer (87% in the mineral part), with 77% in the active



layer (23% in upper permafrost). From a SOC 0–100 cm storage perspective, the alluvial fan is the most important landform, occupying 15% of the area with a mean SOC storage of 21.3 kg C m$^{-2}$ and holding ~ 60% of the total SOC 0–100 cm in the study area (Fig. 2). In contrast, landforms at higher elevation (bedrock, allochthonous weathered bedrock, solifluction sheets, and lateral/end moraine) occupy 70% of the study area but store only 15% of the total SOC 0–100 cm stocks, with mean SOC ranges between 0–2 kg C m$^{-2}$.

This study provides first SOC estimates for the second and third meter depth intervals. However, estimates for deeper layers are based on fewer sites (see subsection 3.2). Mean SOC storage from 100 to 200 cm depth decreased with 65% compared to the top meter to 1.7 kg C m$^{-2}$. The 'alluvial fan class' is the dominant landform also regarding SOC at this depth, with 8.6 kg C m$^{-2}$, contributing with 76% to the total SOC in the study area (Table 2 and Fig. 2). From 200 to 300 cm depth we estimated an additional SOC storage of 0.69 kg C m$^{-2}$, with 71% of the total SOC located in alluvial fans (Table 2 and Fig. 2). Thus, the estimated mean SOC storage for the top 300 cm soil depth is 7.2 kg C m$^{-2}$. However, two of our sampling sites (alluvial fan and delta) had fine-grained deposits that were deeper than 300 cm indicating that buried SOC is present beneath 300 cm depth (Cable et al., 2017). Additionally, Gilbert et al. (2017) report low C densities from two cores relatively close to the coast with 7 and 11 m thick deltaic sediments overlying glacial sediments. However, the areal extent and the average sediment thickness of these deposits are not known.

The mean soil TN storage in the 0–100 cm depth interval in the Zackenberg study area is 0.28 kg TN m$^{-2}$ according to the geomorphological upscaling. The highest values are found in the alluvial fans including the wetlands, with a mean storage of 1.1 kg N m$^{-2}$ in alluvial fans and up to 2.9 kg N m$^{-2}$ in bogs (Table 2). However, because of the small spatial extent occupied by wetlands their contribution to the total TN storage is only c. 2.5%, while alluvial fans are storing 59% of the soil TN at 0–100 cm in the active layer and top permafrost in the study area.

### 5 Discussion

This study presents new estimates of total storage and landscape partitioning of SOC and soil TN in the Zackenberg study area based on detailed geomorphological map upscaling. In comparison with the previous land cover classification (LCC) approach performed for the same area (Palmtag et al., 2015), the geomorphologically-based upscaling shows a 42% reduced weighed mean SOC 0–100 cm storage from 8.3 (± 1.8 CI) to 4.8 (± 1.0 CI) kg C m$^{-2}$. While the SOC 0–100 cm estimates for the organic and permafrost layers deviated little in comparison with the estimates of the previous LCC approach, most deviation occurred within the mineral active layer. Likewise, the mean soil TN storage at 0–100 cm depth decreased with 44% from 0.50 kg (± 0.1 CI) to 0.28 (± 0.1 CI) kg TN m$^{-2}$.

This relatively large difference in SOC (and TN) 0–100 cm storage is mainly due to the importance of geomorphological processes in redistributing sediments in mountainous areas, which is to some extent neglected when using LCCs primarily based on vegetation cover



classification from satellite observations. For example, in the LCC-upscaling (Palmtag et al., 2015) the SOC-rich vegetated classes 'grasslands' and, to a lesser degree, 'fens' occupied relatively large proportions of the total study area (20% and 3% coverage, respectively). These included areas on slopes at mid-elevation with patchy grassland cover and wet areas along

streambeds. However, most of the pedons for these classes were located in the foothills and central valley characterized by higher SOC and TN storage. This resulted in a pedon dataset that was not truly representative for its thematic class and a high-biased mean SOC and TN storage was applied to relatively large areas. The GLC-approach in the current study better identifies areas of high SOC and TN storage in depositional environments such as alluvial fans

(including wetlands on alluvial fans) occupying more limited proportions of the total study area (15% and 0.4%, respectively). Therefore, the substantial decrease of SOC and TN has occurred because the areal extent of the SOC and TN-rich vegetated classes (grasslands 20%, *Salix* snow bed 7%, *Dryas* heath 6%, *Cassiope* heath 4% and fen 3%) has been reduced from 40% in the LCC to 22% (alluvial fans 15.4%, delta (raised, relict) 2.5%, fluvial stream bed (relict) 1.2%,

and ground moraine till 2.4%) using the landform upscaling. Also in non-alpine environments, such as the Lena River Delta, geomorphological setting better explained SOC variability than vegetation cover (Siewert et al., 2016).

When comparing the confidence intervals (CI) of the weighed mean SOC 0–100 cm estimates for the entire study area using LCC ($8.3 \pm 1.8$ kg C m$^{-2}$) and GLC ($4.8 \pm 1.0$ kg C m$^{-2}$) there is

an absolute decrease in the uncertainty range, but in relative terms it remains similar (c. 20%, also for TN). Comparable results are obtained for the mean and standard deviations (SD) of the dominant classes in both upscaling approaches (despite a marked difference in sample size), with 'grasslands' (n=6) in the LCC ($19.1 \pm 8.3$ kg C m$^{-2}$) and 'alluvial fans' (n=15) in the GLC ($19.8 \pm 9.3$ kg C m$^{-2}$), or a coefficient of variation of c. 45% in both cases. This might point to

the fact that there is an intrinsic variability in SOC storage within these classes, related to microtopography, surface wetness, plant productivity, SOC burial, coarse fraction content, permafrost table and ground ice volume and type, among others.

The geomorphological approach is particularly important in identifying areas of deep SOC storage related to depositional environments such as alluvial, fluvial and deltaic landforms, for

which the (cryo)stratigraphy (including excess ground ice) should be taken into account. Alluvial fans can consist of up to several meters ($\leq 4$ m) thick fine-grained laminated deposits accumulated during the Holocene due to the downslope sediment transport of materials and their subsequent deposition in the foothills by nivation processes (Christiansen, 1998). Intercalated SOC-enriched layers up to 8600 cal yr BP old (Cable et al., 2017) indicate the

repeated burial of stable vegetated surfaces and/or organic material eroded by nival meltwater higher up. Relict deltaic areas in the lower central valley are another area of deep SOC storage, with deltaic deposits OSL-dated to 11-13 ka reaching depths of 7–11 m overlying a glacial till unit (Gilbert et al., 2017).

This study includes the first SOC storage estimates down to 300 cm depth with quantitative

uncertainty ranges for the Zackenberg study area. We estimate that $1.7 \pm 0.8$ kg C m$^{-2}$ and 0.7





± 0.4 kg C m⁻² is being stored in the second and third meter of deposits, respectively. This is a considerable amount, representing an additional 50% SOC compared to the storage in the top meter, mainly located in alluvial fan deposits. Deltaic deposits further contribute to the 100–300 cm SOC storage but locally reach depths of at least 11 m. The low carbon densities of

these deepest deposits are similar to those reported for the 100–300 cm depth interval (Cable et al., 2017). However, their areal extent and average depth is unknown, which does not permit an accurate calculation of their total SOC storage.

## 6 Conclusions

This study presents new additional sampling sites and improved estimates of SOC and TN

from 0 to 100 cm depth in Zackenberg (NE Greenland), based on upscaling using geomorphological landforms. Moreover, we report first SOC estimates to a depth of 300 cm. The updated weighed landscape-level mean SOC 0–100 cm storage in the Zackenberg study area based on geomorphological mapping is 42% less than previously reported, when using a LCC upscaling approach. The new mean SOC estimate is 4.8 kg C m⁻² to 100 cm depth,

compared to the original estimate of 8.3 kg C m⁻². A previous areal overestimate of SOC-rich vegetated land cover classes on slopes was the main reason for this large difference in SOC storage. Downslope creep constantly transport material downslope resulting in relatively shallow soil depths and low SOC storage. Slope materials have accumulated at the foot of the slopes in alluvial fans during the entire Holocene, primarily by nivation processes creating

thick, fine-grained deposits with buried SOC-enriched layers throughout their depth. Whereas the LCC recognized vegetated SOC-rich classes on the slopes, in the foothills and in the central valley, the GLC upscaling more correctly restricts the SOC-rich classes to areas with deposition. The use of LCC upscaling in these mountainous settings can introduce large uncertainties since it is based on recent land cover and vegetation only that do not necessarily

reflect the long-term geomorphic processes leading to SOC burial. To the contrary, the landform-based approach identifies hotspots of SOC burial in the landscape such as alluvial fans and (to a minor extent) deltas. The GLC approach is, therefore, also highly relevant when identifying areas of deep carbon storage (between 100–300 cm, and more). Our deep pedon dataset indicates that an additional 1.7 and 0.7 kg C m⁻² are stored in the second and third meter

soil depth, respectively. Deltaic deposits extend below 300 cm depth, which implies that there are additional SOC stocks at greater depth. The results emphasize the importance of geomorphology, rather than land cover, controlling SOC storage in high relief permafrost environments.

**Acknowledgements**

Fieldwork in Zackenberg was financed through the Nordforsk Nordic Centre of Excellence DEFROST project (grant number 23001), the EU FP7 PAGE21 project (grant number 282700)



and the ESF CryoCARB project (Swedish Research Council project to P. Kuhry). Additional funding for fieldwork in 2013 and subsequent core analyses were supported by the University Centre in Svalbard (UNIS) and the Centre for Permafrost (CENPERM) at the University of Copenhagen, funded by the Danish National Research Foundation (CENPERM DNRF100).

We gratefully acknowledge the support with drilling and core analyses by co-leader Professor Bo Elberling (CENPERM, University of Copenhagen) and students participating in the UNIS course AG-833 'High Arctic Permafrost Landscape Dynamics in Svalbard and Greenland' in 2013, funded by the University Centre in Svalbard (UNIS), the Norden Perma-Nordnet project and the Centre for Permafrost (CENPERM), University of Copenhagen. We would like to

thank the GeoBasis Programme of the Zackenberg research station for providing data (overview map and DEM data).

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



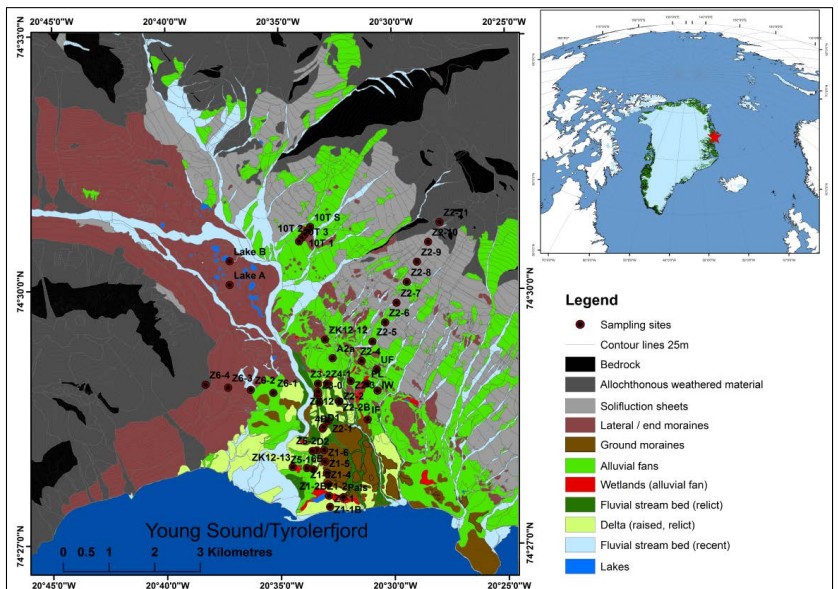

**Figure 1. Simplified geomorphological landform map of the Zackenberg study area (based on Cable et al., 2017). Top right: location of Zackenberg in NE Greenland.**



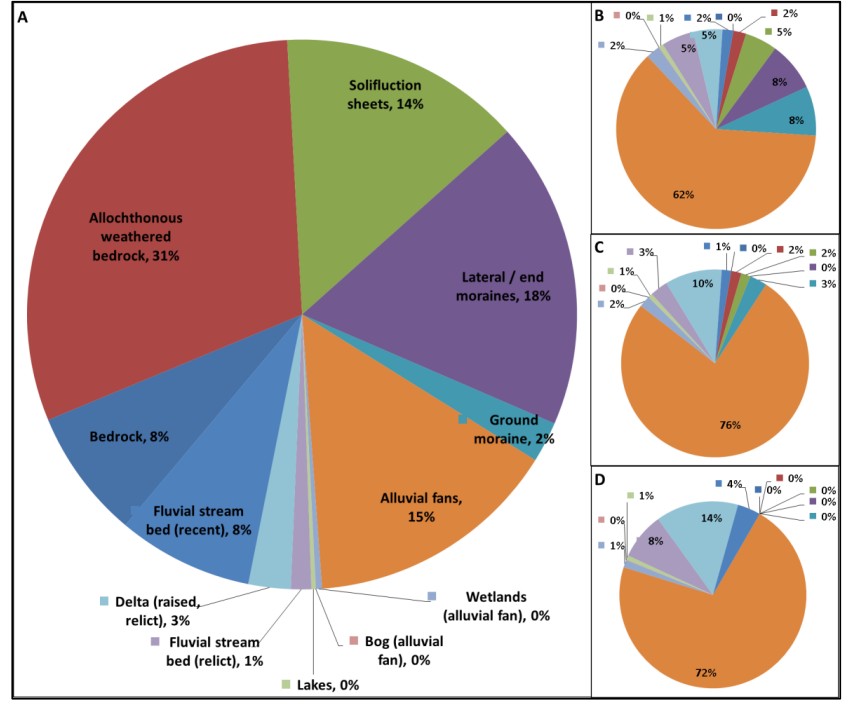

**Figure 2. Proportional contribution of each landform to: A) Areal coverage of the total study area;**
5  **B) total SOC storage for 0–100 cm; C) total SOC storage for 100–200 cm; and D) total SOC storage**
**for 200–300 cm. Landform and colors from chart A apply also to charts B-D.**



**Table 1. Geomorphological landforms and their proportion of total surface area as presented in Cable et al. (2017), and the amalgamated larger landform classes used in this study (excluding Young Sound).**

| Geomorphological landforms | Area % | Overall geomorphological classes | Area % |
|---|---|---|---|
| Bedrock, exposed | 7.62 | Bedrock | 7.62 |
| Bedrock, weathered, allochthonous | 26.0 | Allochthonous weathered bedrock | 30.4 |
| Debris cone, debris fan | 2.50 | | |
| Rock glacier | 0.34 | | |
| Perennial snow patch[1] | 1.53 | | |
| Slide scar | 0.03 | Solifluction sheets | 14.3 |
| Solifluction sheet, solifluction lobe | 14.3 | | |
| Moraine, ice-cored | 4.78 | Lateral / end moraines | 18.1 |
| Lateral moraine | 14.3 | | |
| Ground moraine | 2.39 | Ground moraine | 2.39 |
| Alluvial fan, coarse-grained, inactive | 0.47 | Alluvial fans | 15.0 |
| Nivation hollow | 3.55 | | |
| Alluvial fan | 10.9 | | |
| Wetland[2] | 0.38 | Wetlands (alluvial fan)[2] | 0.38 |
| River, channel, relict | 1.20 | Fluvial stream bed (relict) | 1.20 |
| Delta, relict | 2.43 | Delta (raised, relict) | 2.48 |
| Sandbar, relict | 0.05 | | |
| River, channel | 3.48 | Fluvial stream bed (recent) | 7.97 |
| Braided river | 1.94 | | |
| Alluvial fan, coarse-grained | 1.32 | | |
| Delta pre-recent | 0.54 | | |
| Spit | 0.05 | | |
| Beach | 0.12 | | |
| Delta, active | 0.53 | | |
| Lacustrine | 0.00 | Lakes | 0.25 |
| Lake | 0.25 | | |
| Total | 100 | | 100 |

[1] Snow patches found at higher elevations among the other landform units belonging to the same amalgamated landform
[2] Wetland areas developed on top of alluvial fan deposits





Table 2. Mean SOC and TN estimates (kg m⁻²) for the Zackenberg study area by landforms.

| Landforms | Type of deposits | Area % | n sites | Mean ± SD SOC/TN storage | | | | | | |
|---|---|---|---|---|---|---|---|---|---|---|
| | | | | Organic layer kg C m⁻² | 0–30 cm kg C m⁻² | 0–100 cm kg C m⁻² | Permafrost in 0–100 cm kg C m⁻² | 100–200 cm kg C m⁻² | 200–300 cm kg C m⁻² | 0–100 cm kg N m⁻² |
| Bedrock | | 7.62 | 1 | 0.0 | 0.0 | 0.0 | 0.0 | 0.0 | 0.0 | 0.0 |
| Allochthonous weathered bedrock | colluvial deposits | 30.4 | 2 | 0.0 ± 0.0 | 0.1 ± 0.1 | 0.3 ± 0.4 | 0.2 ± 0.3 | 0.1 ± 0.1 | 0.0 | 0.0 |
| Solifluction sheets | colluvial deposits | 14.3 | 3 | 0.1 ± 0.6 | 0.1 ± 0.4 | 1.7 ± 0.9 | 0.0 | 0.2 ± 0.3 | 0.0 | 0.1 ± 0.1 |
| Lateral / end moraines | glacial deposits | 18.1 | 4 | 0.1 ± 0.1 | 0.9 ± 1.1 | 2.1 ± 3.2 | 0.0 | 0.0 | 0.0 | 0.1 ± 0.1 |
| Ground moraine | glacial deposits | 2.39 | 3 | 1.6 ± 0.2 | 6.7 ± 3.1 | 15.9 ± 4.0 | 1.7 ± 2.8 | 2.1 ± 1.8 | 0.0 | 1.1 ± 0.3 |
| Alluvial fans | alluvial deposits | 15.0 | 15 | 2.5 ± 3.0 | 7.3 ± 3.6 | 19.8 ± 9.3 | 3.9 ± 4.1 | 8.6 ± 10.0 | 3.3 ± 5.8 | 1.1 ± 0.6 |
| Wetlands (alluvial fan) | peat/alluvial deposits | 0.36 | 5 | 16.0 ± 17.7 | 11.0 ± 3.4 | 29.8 ± 12.5 | 14.4 ± 10.1 | 9.1 ± 10.0 | 2.4 ± 2.2 | 1.9 ± 0.7 |
| Bog (alluvial fan) | peat/alluvial deposits | 0.01 | 3 | 36.5 ± 8.8 | 16.0 ± 9.6 | 42.7 ± 7.0 | 24.6 ± 17.2 | 6.2 ± 2.6 | 6.2 ± 2.6 | 2.9 ± 0.5 |
| Lakes | lacustrine deposits | 0.25 | 4 | 11.4 ± 1.3 | 5.6 ± 2.2 | 14.5 ± 2.8 | 1.0 ± 1.2 | 5.0 ± 2.5 | 2.1 ± 1.8 | 0.4 ± 0.0 |
| Fluvial stream bed (relict) | fluvial deposits | 1.20 | 2 | 2.3 ± 0.8 | 6.2 ± 0.2 | 20.9 ± 4.5 | 8.4 ± 6.3 | 4.4 ± 1.3 | 1.4 ± 1.7 | 1.4 ± 0.4 |
| Delta (raised, relict) | deltaic deposits | 2.48 | 4 | 1.2 ± 0.8 | 4.4 ± 1.6 | 9.1 ± 3.2 | 2.2 ± 2.3 | 6.7 ± 4.4 | 4.0 ± 1.4 | 0.7 ± 0.3 |
| Fluvial stream bed (recent) | fluvial deposits | 7.97 | 2 | 0.1 ± 0.2 | 0.8 ± 0.8 | 1.0 ± 0.5 | 0.0 | 0.3 ± 0.5 | 0.3 ± 0.5 | 0.0 |
| Mean ± CI SOC/TN storage (weighed by landform proportion) | | 100 | 48 | 0.6 ± 0.3 | 1.9 ± 0.4 | 4.8 ± 1.0 | 1.1 ± 0.4 | 1.7 ± 0.8 | 0.7 ± 0.4 | 0.3 ± 0.1 |