# Peer review of "Landform partitioning and estimates of deep storage of soil organic matter in Zackenberg, Greenland"

_The Cryosphere, 2017_

## Referee Comment (RC1) · Anonymous Referee #1 · 27 Dec 2017

This well-written manuscript depicts a research project studying the carbon and nitrogen stores by applying landform partition in a deglaciated valley in NE Greenland. This study involved well-planned soils sampling scheme, well-defined methodology. The most exciting part of this study is that it was carried out in exactly the same area by other researchers using different approach. Thus, the researchers in this study be able to compare the validity and limitations of the GLC and LCC approaches in assessing the C & N stores in such a high relief area affected by permafrost. The results from this study would contribute to our understanding of the relationships between C & N stores

and landscape dynamics in alpine environments and recently deglaciated areas.

References listed are adequate, all figures and tables well-done.

The quality of the manuscript and the scientific merit of the study warrant its worthiness to be accepted by Cryosphere.

There are some specific comments for the authors to consider.

P 4. L 22. "time or". Isn't time part of the logistical problem?

P 6. L 7-15. Materials in 2 glacial moraines (lateral and end) deeper than 100 was considered as "till". Since soils sampled from 0-100 cm in both landform formed in moraine, the parent material of these soils is moraine. Then, why call the portion below 100 cm till?

P 6. L. 31-32. C stores reported for 0-300 cm in small lakes. Not clear if the whole 300 cm includes both water and sediment?

P 7. L 37. "importance of". Would it be better say "effects of" or "important role of"?

P 8. L 25-27. Consider change "surface wetness" to "drainage" and permafrost table" to "depth of permafrost" or "active layer thickness".

---

## Referee Comment (RC2) · Anonymous Referee #2 · 3 Jan 2018

This paper addresses an important research filed and is well written and structured. Despite some weaknesses in the discussion (what is the bigger picture, what could this great dataset be used for...) I recommend an acceptance of the manuscript after including some revisions. Please find my detailed comments below.

TC review criteria

1. Does the paper address relevant scientific questions within the scope of TC? The paper addressed a crucial topic of permafrost research: The inventory of organic mat-

ter, which is needed for climate-response assessments of the Arctic

2. Does the paper present novel concepts, ideas, tools, or data? The paper provides updated numbers on organic matter for the Zackenberg area. The methods are based on a well-tried approach following Hugelius et al (2013, 2014) as well as the geomorphological map by Cable et al. (2017). Thus, there are no new concepts, ideas, or tools, but new data with improved upscaling techniques.

3. Are substantial conclusions reached? Yes, there is an improves estimate on the organic matter inventory

4. Are the scientific methods and assumptions valid and clearly outlined? Yes

5. Are the results sufficient to support the interpretations and conclusions? Yes

6. Is the description of experiments and calculations sufficiently complete and precise to allow their reproduction by fellow scientists (traceability of results)? Yes

7. Do the authors give proper credit to related work and clearly indicate their own new/original contribution? Yes

8. Does the title clearly reflect the contents of the paper? The title is very long; I recommend shortening and strengthening it. But yes, in this version the title reflect the paper in very detail

9. Does the abstract provide a concise and complete summary? Yes

10. Is the overall presentation well-structured and clear? Yes, the manuscript is well written and presented.

11. Is the language fluent and precise? Yes

12. Are mathematical formulae, symbols, abbreviations, and units correctly defined and used? Yes

13. Should any parts of the paper (text, formulae, figures, tables) be clarified, reduced,

combined, or eliminated? See detailed comments below: There is some potential for another figure like changing a table to boxplots or a figure adressing the discussion

14. Are the number and quality of references appropriate? In general yes, except for the discussion

15. Is the amount and quality of supplementary material appropriate? There is no supplementary material attached

Detailed comments Page 1 L1 (title): Very long title, but the nitrogen is missing. What about "Deep storage of organic matter in Zackenberg, Greenland". The methodical approach ("geomorphological landform approach") is not needed here

L13: Please add an introductory sentence before stating the paper aims. General comment: are all the included samples soils in the sense of soul science? If yes, SOC is an appropriate abbreviation, otherwise please change to OC. Please add to the abstract a short discussion sentence including "what is deeper than 3 m" (you did this in the paper text already for fan/delta) and "how representative" could this study area be for Greenland/or other Arctic areas (missing right now)

L30: Please add a reference here

L31: Please concretize "most regions". Permafrost regions, Arctic regions? Despite IPCC, please add a primary reference here

Page 2 L4: As you know first assessment was not Tarnocai et al 2009, there have been a lot more before, like Post et al 1982, Tarnocai 2003, than Zimov 2006

L16: add a dot after . . .2016)

L29 and following: I like the concise presentation of the specific aims. Please be consistent and use this order and number for the conclusion sections

L36: please delete the c. or introduce this abbreviation for the coordinates.

L38: Please add the rain/snow percentages of the precipitation (if available)

Page 3 L15: Please add the GPS uncertainty here

L19: Not the motor head, but the core barrel system is of relevance here. Please add this information here.

L32: How many samples were measured for TOC/TN. TOC and TN in one run and same device? Or did you decarbonise the samples before to get rid of the TIC? Did you calibrate your 950 TIC measurements as well?

Page 4 Line 9: Please add the units to the formula, this would explain the percentage /100 and g to kg as well as the $cm^3$ to $m^2$ conversion factor

L16: Please explain the mentioned "data limitations"

L22: please delete "time or"

Page 5 L3: Is your data Gaussian distributed? Did you test this (e.g. Shapiro-Wilk test). Why not using more robust median and interquartile ranges? Please explain and justify your decisions here.

L16: Same for CI, which requires Gaussian distributed data

General comments results section: I understand you n is the number of sites, but could you add a n for the number of measurements as well? Is the percentage a result of Cable 2017 or this study? I recommend transforming the major information of Table 2 into a boxplot figure (which is allowed for a n<5 for n (measurements, nit sites) and include table 2 into a supplementary chapter. Is the data available (embargoed, e.g. PANGAEA). Then you can add a doi of your data.

Page 7 L7: First estimated for Zackenberg or Greenland or similar landscapes?

L24: please delete c., which is not coordinates for which this abbreviation was used before

General comments discussion: Please include and discuss a number of how much is perennially frozen/seasonally unfrozen of you carbon and nitrogen stocks Please include comparisons of your and other comparable case studies, which you all know because of author overlap (e.g. Fuchs et al. Biogeosciences Discussions, Hugelius et al. 2009 etc.) To my opinion 5 references (4 actually, as Palmtag 2015 is mentioned twice) for a discussion are not enough. I do not care about the number of references, but this shows that the authors should discuss their findings a bit bigger context. Frozen/unfrozen percentages, what are the consequences for the carbon pool –> modelling... any back on the envelope calculation of the <3m pool of the fans etc. Discuss the satellite product comparisons (Bartsch and other ESA DUE or GlobPermafrost-related publications). Rough discussion of carbon qualities (using C/N) and compare this to literature concepts (for C/N Schaedel et al 2014) Maybe an additional "discussion figure" (right now just 1 intro and 1 results figure) would be helpful

Page 8 L18: Keep using introduced abbreviations (here CI) and do not introduce them twice.

Page 9 L6: Making an rough estimation by stating the assumption of a <3m pool could be a nice first guess for future work

General comments for the conclusions: Please repeat the paper's aims here and answer this in the same order like in introduction. Please include your nitrogen calculations here as well.

L9: "new additional" sounds strange

Figures and Table: Figure 1: hard to read the site labels. Maybe a hillshade and transparent colours could improve the geomorphological understanding for the reader. According to figure 2: use A and B instead of "top right"

Figure 2: A is redundant with Table 1, right? An option could be deleting a and put the

spotlight on B-C

Table 2: pleas add n measurements to the table. Moreover switching from CI (in the manuscript text) to SD could be puzzling for the readers

---

## Author Comment (AC1) · 21 Feb 2018

We thank the referee for their constructive comments of this manuscript which helped to improve this manuscript.

Comment: P 4. L 22. "time or". Isn't time part of the logistical problem? Response: We agree that it is part of logistical problems and remove "time or" from the text.

Comment: P 6. L 7-15. Materials in 2 glacial moraines (lateral and end) deeper than 100 was considered as "till". Since soils sampled from 0-100 cm in both landform formed in moraine, the parent material of these soils is moraine. Then, why call the portion below 100 cm till? Response: We refer to the deeper deposits as till from a sedimentological perspective and not as a landform, based on the analyzed coring material which was unsorted glacial sediment

Comment: P 6. L. 31-32. C stores reported for 0-300 cm in small lakes. Not clear if the whole 300 cm includes both water and sediment? Response: This is a good point so we added additional clarification in the text "from sediment surface".

Comment: P 7. L 37. "importance of". Would it be better say "effects of" or "important role of"? Response: We changed as suggested to "important role of".

Please also note the supplement to this comment:
https://www.the-cryosphere-discuss.net/tc-2017-255/tc-2017-255-AC1-supplement.pdf

[Figure]

**Supplement:**

[revised manuscript text omitted]

---

## Author Comment (AC2) · 21 Feb 2018

We are grateful for the detailed and constructive comments by the reviewer which were helpful in improving the scientific quality and clarity of this manuscript.

Anonymous Referee #2

Comment: Page 1 L1 (title): Very long title, but the nitrogen is missing. What about

"Deep storage of organic matter in Zackenberg, Greenland". The methodical approach ("geomorphological landform approach") is not needed here.

Response: We agree that the title was very long and used partly used the suggestion by the reviewer. But since the actual aim of this paper is the used approach, we think that landform partitioning should be still part of the title.

Comment: L13: Please add an introductory sentence before stating the paper aims. General comment: are all the included samples soils in the sense of soil science? If yes, SOC is an appropriate abbreviation, otherwise please change to OC. Please add to the abstract a short discussion sentence including "what is deeper than 3 m" (you did this in the paper text already for fan/delta) and "how representative" could this study area be for Greenland/or other Arctic areas (missing right now).

Response: As asked we added an introductory sentence: "Soils in the northern high latitudes are a key component in the global carbon cycle, with potential feedbacks on climate." We also changed for the deeper deposits the abbreviation to OC and added which type of sediment it is "alluvial and deltaic deposits". We also added in the discussion section rough estimations on SOC storage in deposits below 3m.

Comment: L30: Please add a reference here Response: Thank you for your comment. We have added additional reference of Schuur et al., 2015.

Comment: L31: Please concretize "most regions". Permafrost regions, Arctic regions? Despite IPCC, please add a primary reference here.

Response: Thank you again for your comment. Primary reference added and the regions are now concretized with: "northern circumpolar regions".

Comment: Page 2 L4: As you know first assessment was not Tarnocai et al 2009, there have been a lot more before, like Post et al 1982, Tarnocai 2003, than Zimov 2006.

Response: We agree and thankful for the comment. This part is now corrected with the proper references regarding the first assessment: "In 2009, Tarnocai et al. linked

circumpolar SOC data (e.g. Kuhry et al., 2002; Zimov et al., 2006; Tarnocai et al, 2007; Ping et al, 2008) and presented a new total estimate of 1674 Petagram C (PgC) stored in soils and deep deposits of the northern circumpolar permafrost region.".

Comment: L16: add a dot after 2016)... Response: Typo corrected and dot added.

Comment: L29 and following: I like the concise presentation of the specific aims. Please be consistent and use this order and number for the conclusion sections

Response: The conclusion section is now restructured and the aims are following the same order.

Comment: L36: please delete the c. or introduce this abbreviation for the coordinates. Response: Agree that c is unnecessary and gladly removed from the text.

Comment: L38: Please add the rain/snow percentages of the precipitation (if available) Response: Asked information on rain/snow added to the site description.

Comment: Page 3 L15: Please add the GPS uncertainty here Response: Approximate GPS uncertainty added.

Comment: L19: Not the motor head, but the core barrel system is of relevance here. Please add this information here.

Response: As asked we added the additional information on the core barrel in the methods section.

Comment: L32: How many samples were measured for TOC/TN. TOC and TN in one run and same device? Or did you decarbonise the samples before to get rid of the TIC? Did you calibrate your 950 TIC measurements as well?

Response: Added and explained in more detail the questions raised by the reviewer #2 on the number of samples, device and sample preparation.

Comment: Page 4 Line 9: Please add the units to the formula, this would explain the
percentage /100 and g to kg as well as the cm3 to m2 conversion factor

Response: The units are explained just above the equation which is a common way to do it. It seems very unnecessary to repeat it again.

Comment: L16: Please explain the mentioned "data limitations"

Response: The text part is now reworked explaining the data limitation for the lack of deeper TN values.

Comment: L22: please delete "time or" Response: Same comment as from reviewer 1, so we removed time from the text.

Comment: Page 5 L3: Is your data Gaussian distributed? Did you test this (e.g. Shapiro-Wilk test). Why not using more robust median and interquartile ranges? Please explain and justify your decisions here.

Response: Thank you for your comment. Yes, in most cases there are too few sites to do a proper test for the normal distribution. However, yes I did test this using Shapiro-Wilk for example in the alluvial fans based on 15 sites. Regarding the usage of standard deviation: We follow a standard protocoll used by our group and others. For that reason and for comparability between sites we use SD.

Comment: L16: Same for CI, which requires Gaussian distributed data Response: See response above, where tested data is normal distributed.

General comments results section: I understand you n is the number of sites, but could you add a n for the number of measurements as well? Is the percentage a result of Cable 2017 or this study? I recommend transforming the major information of Table 2 into a boxplot figure (which is allowed for a n<5 for n (measurements, nit sites) and include table 2 into a supplementary chapter. Is the data available (embargoed, e.g. PANGAEA). Then you can add a doi of your data.

Response: As asked by the reviewer, we added the number of measurements in Table

2. Question regarding the percentage landform cover. As stated in the text, we amalgamated small classes from Cable et al., where we didn't have any sampling pedons into larger landform classes. However we use the same aerial extend of the study area. Table 1 shows the geomorphological landforms and their proportion of total surface area in Cable et al., into the amalgamated classes. As stated in a former response, we would like to keep the table with all the data as it is which enables a direct comparison with other sites dealing with SOC storage. Also, the data is not yet available in e.g. PANGAEA but is planned in the future.

Comment: Page 7 L7: First estimated for Zackenberg or Greenland or similar landscapes? Response: Additional information added to clarify this point.

Comment: L24: please delete c., which is not coordinates for which this abbreviation was used before Response: c which was used for circa now removed from several places.

General comments discussion: Please include and discuss a number of how much is perennially frozen/seasonally unfrozen of you carbon and nitrogen stocks. Please include comparisons of your and other comparable case studies, which you all know because of author overlap (e.g. Fuchs et al. Biogeosciences, Discussions, Hugelius et al. 2009, etc.) To my opinion 5 references (4 actually, as Palmtag 2015 is mentioned twice) for a discussion are not enough. I do not care about the number of references, but this shows that the authors should discuss their findings a bit bigger context. Frozen/unfrozen percentages, what are the consequences for the carbon pool –> modelling...any back on the envelope calculation of the <3m pool of the fans etc. Discuss the satellite product comparisons (Bartsch and other ESA DUE or GlobPermafrost-related publications). Rough discussion of carbon qualities (using C/N) and compare this to literature concepts (for C/N Schaedel et al 2014) Maybe an additional "discussion figure" (right now just 1 intro and 1 results figure) would be helpful.

Response: Thank you for these comments. Several more references were now added

and their findings discussed and compared to our results. We also included comments on the perennially frozen proportion. However, there are too few sites at all available with comparable mountainous permafrost environment and to my knowledge none of those which deals with carbon stocks deeper than 1m of depth. We also included the publication from Bartsch et al. using SOC estimated based on synthetic aperture radar. But since the soil penetration is only a few cm, it has the similar problem as LCC and cannot capture the long-term depositional history. And we believe that a discussion of carbon qualities is beyond the scope of this paper.

Comment: Page 8 L18: Keep using introduced abbreviations (here CI) and do not introduce them twice. Response: Mistake corrected and the secondary introduction of abbreviation removed.

Comment: Page 9 L6: Making an rough estimation by stating the assumption of a <3m pool could be a nice first guess for future work Response: We added a rough estimation for SOC storage in deltaic deposits below 3m.

General comments for the conclusions: Please repeat the paper's aims here and answer this in the same order like in introduction. Please include your nitrogen calculations here as well. Response: The comment implemented and the aims reordered following the introductions order. Also, the nitrogen was added in this section.

Comment: L9: "new additional" sounds strange Response: The "additional" is now removed from the text.

Comment: Figures and Table: Figure 1: hard to read the site labels. Maybe a hillshade and transparent colours could improve the geomorphological understanding for the reader. According to figure 2: use A and B instead of "top right" Response: Both comments implemented.

Comment: Figure 2: A is redundant with Table 1, right? An option could be deleting a and put the spotlight on B-C.

Response: Figure 2 A is redundant with Table 1 but we would still like to keep is as it, since it illustrates graphically the proportional contribution of each landform in comparison to the SOC storage for each landform.

Comment: Table 2: please add n measurements to the table. Moreover switching from CI (in the manuscript text) to SD could be puzzling for the readers.

Response: As suggested we added in the table 2 the number of samples measured for each landform. Regarding the use of SD and CI: In the manuscript text and Table 2 it is clearly stated if we used SD (only for the landform means) or CI which was used for the whole Zackenberg area. The decision to use CI is because it account for the relative spatial coverage, storage variability and degree of replication of each upscaling class and therefore much more confident.

Please also note the supplement to this comment:
https://www.the-cryosphere-discuss.net/tc-2017-255/tc-2017-255-AC2-supplement.pdf

**Supplement:**

[revised manuscript text omitted]

---

## Editor Comment (EC1) · J. Boike (Editor) · 27 Feb 2018

Dear authors,

many thanks for your replies and your revised version. In my editorial letter I had asked you address the initial comments provided by guest editor Scott Lamoreux.

I attach the comments below again.

Please provide a third point to point reply to these comments, and potentially a revised

[Figure]

paper.

Many thanks, Julia

Comments

In the discussion, you really don't discuss the TN. The TN appears to be an afterthought in the analysis, but I suspect many readers would be greatly interested by this data and spatial variations.

I would also suggest presenting some of your data as a figure showing SOC and TN trends with depth. You could provide this for representative sites. This would be a simple but effective figure.

Specific comments:

-Refer to TN consistently through the text when referring to SOC as appropriate.

-p 2 line 34 and elsewhere. Use of Arctic as a proper noun (capitalized). An argument for correct usage of "Arctic" is made by Kingsley (2005, Arctic). I suggest you apply this throughout, and consider "High Arctic" in the same manner.

-p3 line 24 give temperature of initial oven drying.

-p3 line 30 a fifth order polynomial is quite a complex fitting, you should consider showing this relationship as a figure or providing the polynomial equation.

-p4 line 4 Define DBD and other terms with first usage.

-p4 line 30 Give dates of aerial imagery

-p5 line 13 Equation reference in text should be (3)

-p5 line 22 remove "software", spell out Microsoft

-p6 line 30 suggest "substantially" rather than significantly as there is no supporting determination
-p6 line 34 you indicate active layer but have not provided depths for this. I suggest mentioning typical or range of active layer depth in the Study Area section. I can infer from this statement active layer is 77 cm?

-p7 line 3 change with to "by"

-p7 line 15 add "to" after according

-p9 line 1 I suggest removing "additional sampling sites". The major contribution here is the refinement of the SOC estimates.

-p9 lines 24-6 This statement seems to indicate that geomorphology is more important than land cover in determining SOC stocks. I think you might wish to elaborate on this statement to avoid misinterpretation. I would argue that you have shown that geomorphology and land cover are important.

-p11 line 5 I think you need "Intergovernmental" at the beginning of this reference.

-p11 line 26 I would indicate this as the USDA for authorship.

-Figure 1: As I look at this and consider your results, I wonder if you could comment on the SOC and the relative age of the various geomorphic units. Some are clearly early Holocene while others have developed during the Holocene, and it would be helpful to link SOC to this, even if briefly (and with limited or no dating). Moraines, as an example, are older, while fluvial and alluvial features are more recent. This connects to your results related to deeper SOC as well, especially on slopes.

-Figure 1: the text on the water would be more visible as white font.

-Figure 2: I suggest that this figure would be improved by making all four pie charts the same size. The land classification is too large for the information it provides.